# Exact Recovery of Stochastic Block Model by Ising Model

**DOI:** 10.3390/e23010065

**Published:** 2021-01-02

**Authors:** Feng Zhao, Min Ye, Shao-Lun Huang

**Affiliations:** 1Department of Electronics, Tsinghua University, Beijing 100084, China; zhaof17@mails.tsinghua.edu.cn; 2Tsinghua Berkeley Shenzhen Institute, Berkeley, CA 94704, USA; yeemmi@sz.tsinghua.edu.cn

**Keywords:** stochastic block model, exact recovery, Ising model, maximum likelihood, metropolis sampling

## Abstract

In this paper, we study the phase transition property of an Ising model defined on a special random graph—the stochastic block model (SBM). Based on the Ising model, we propose a stochastic estimator to achieve the exact recovery for the SBM. The stochastic algorithm can be transformed into an optimization problem, which includes the special case of maximum likelihood and maximum modularity. Additionally, we give an unbiased convergent estimator for the model parameters of the SBM, which can be computed in constant time. Finally, we use metropolis sampling to realize the stochastic estimator and verify the phase transition phenomenon thfough experiments.

## 1. Introduction

In network analysis, community detection consists in inferring the group of vertices that are more densely connected in a graph [1]. It has been used in many domains, such as recommendation systems [2], task allocation in distributed computing [3], gene expressions [4], and so on. The stochastic block model (SBM) is one of the most commonly used statistical models for community detection problems [5,6]. It provides a benchmark artificial dataset to evaluate different community detection algorithms and inspires the design of many algorithms for community detection tasks. These algorithms, such as semi-definite relaxation, spectral clustering, and label propagation, not only have theoretical guarantees when applied to the SBM, but also perform well on datasets without the SBM assumption. The study of the theoretical guarantee of the SBM model can be divided between the problem of exact recovery and that of partial recovery. Exact recovery requires that the estimated community should be exactly the same as the underlining community structure of the SBM whereas partial recovery expects the ratio of misclassified nodes to be as small as possible. For both cases, the asymptotic behavior of the detection error is analyzed when the scale of the graph tends to infinity. There are already some well-known results for the exact recovery problem on the SBM. To name but a few, Abbe and Mossel established the exact recovery region for a special sparse SBM with two communities [7,8]. Later on, the result was extended to a general SBM with multiple communities [9].

Parameter inference in the SBM is often considered alongside the exact recovery problem. Previous inference methods require the joint estimation of node labels and model parameters [10], which have high complexity since the recovery and inference tasks are done simultaneously. In this article, we will decouple the inference and recovery problems, and propose an unbiased convergent estimator for SBM parameters when the number of communities is known. Once the estimator is obtained, the recovery condition can be checked to determine whether it is possible to recover the labels exactly. Additionally, the estimated parameter will guide the choice of parameters for our proposed stochastic algorithm.

In this article, the exact recovery of the SBM is analyzed by considering the Ising model, which is a probability distribution of node states [11]. We use the terms node states and node labels interchangeably throughout this paper, both of which refer to the membership of the underlining community. The Ising model was originally proposed in statistical mechanics to model the ferromagnetism phenomenon but has wide applications in neuroscience, information theory, and social networks. Among different variants of Ising models, the phase transition property is shared. Phase transition can be generally formulated when some information quantity changes sharply in a small neighborhood of parameters. Based on the random graph generated by an SBM with two underlining communities, the connection of the SBM and the Ising model was first studied by [12]. Our work will extend the existing result to the multiple community case, establish the phase transition property, and give the recovery error an upper bound. The error bounds decay in a polynomially fast rate in different phases. Then we will propose an alternative approach to estimate the labels by finding the Ising state with maximal probability. Compared with sampling from the Ising model directly, we will show that the optimization approach has a sharper error upper bound. Solving the optimization problem is a generalization of maximum likelihood and also has a connection with maximum modularity. Additionally, searching the state with maximal probability could also be done within all balanced partitions. We will show that this constrained search is equivalent to the graph minimum cut problem, and the detection error upper bound for the constrained maximization will also be given.

The exact solution to maximize the probability function or exact sampling from the Ising model is NP-hard. Many polynomial time algorithms have been proposed for approximation purposes. Among these algorithms, simulated annealing performs well and produces a solution that is very close to the true maximal value [13]. On the other hand, in the original Ising model, metropolis sequential sampling is used to generate samples for the Ising model [14]. Simulated annealing can be regarded as metropolis sampling with decreasing temperature. In this article, we will use the metropolis sampling technique to sample from the Ising model defined on the SBM. This approximation enables us to verify the phase transition property of our Ising model numerically.

This paper is organized as follows. Firstly, in Section 3 we introduce the SBM and give an estimator for the parameters of the SBM. Then, in Section 4, our specific Ising model is given and its phase transition property is obtained. Derived from the Ising model, in Section 5, the energy minimization method is introduced, and we establish its connection with maximum likelihood and modularity maximization algorithm. Furthermore, in Section 6, we realize the Ising model using the metropolis algorithm to generate samples. Numerical experiments and conclusion are given lastly to finish this paper.

Throughout this paper, the  community number is denoted by *k*; the random undirected graph *G* is written as G(V,E) with vertex set *V* and edge set *E*; V={1,…,n}=:[n]; the label of each node is $X_{i}$, which is chosen from $W=\{1,\omega ,\ldots ,\omega^{k-1}\}$, and we further require *W* to be a cyclic group with order *k*; $W^{n}$ is the n-ary Cartesian power of *W*; *f* is a permutation function on *W* and is extended to $W^{n}$ in an element-wise manner; $U^{c}$ is the complement set of *U* and |U| is the cardinality of *U*; the set $S_{k}$ is used to represent all permutation functions on *W* and $S_{k}\left(\sigma \right):=\left\{f\left(\sigma \right)|f\in S_{k}\right\}$ for $\sigma \in W^{n}$; the indicator function δ(x,y) is defined as δ(x,y)=1 when x=y, and δ(x,y)=0 when x≠y; f(n)=O(g(n)) if there exists a constant c>0 such that f(n)≤cg(n) for large *n*; f(n)=o(g(n)) holds if $\lim_{n\to \infty }\frac{f(n)}{g(n)}=0$; we define the distance of two vectors as: $\mathrm{dist}\left(\sigma ,\sigma^{\prime }\right)=\left|\left\{i\in \left[n\right]:\sigma_{i}\neq \sigma_{i}^{\prime }\right\}\right|\mathrm{for}\sigma ,\sigma^{\prime }\in W^{n}$ and the distance of a vector to a space $S\subseteq W^{n}$ as $\mathrm{dist}\left(\sigma ,S\right):=\min\{\mathrm{dist}\left(\sigma ,\sigma^{\prime }\right)|\sigma^{\prime }\in S\}$. For example, when n=2 and k=2, $\sigma =\left(1,\omega \right)\in W^{2}$; $\omega^{0}=1$; $\omega \cdot \omega =\omega^{2}=1$; let *f* be a mapping such that f(1)=ω and f(ω)=1, then $f\in S_{2}$ and f(σ)=(ω,1); dist(σ,f(σ))=2; $S_{k}\left(\sigma \right)=\left\{\sigma ,f\left(\sigma \right)\right\}$; and $S_{k}^{c}\left(\sigma \right)=\left\{\left(1,1\right),\left(\omega ,\omega \right)\right\}$.

## 2. Related Works

The classical Ising model is defined on a lattice and confined to two states {±1}. This definition can be extended to a general graph and multiple-state case [15]. In [16], Liu considered the Ising model as defined on a graph generated by sparse SBM and his focus was to compute the log partition function, which was averaged over all random graphs. In [17], an Ising model with a repelling interaction was considered on a fixed graph structure, and the phase transition condition was established, which involves both the attracting and repelling parameters. Our Ising model derives from the work of [12], but we extend their results by considering the error upper bound and multiple-community case.

The exact recovery condition for the SBM can be derived as a special case of many generalized models, such as pairwise measurements [18], minimax rates [19], and side information [20]. The Ising model in this paper provides another way to extend the SBM model and derives the recovery condition. Additionally, the error upper bound for exact recovery of the two-community SBM by constrained maximum likelihood has been obtained in [7]. Compared with previous results, we establish a sharper upper bound for the multiple-community case in this paper.

The connection between maximum modularity and maximal likelihood was investigated in [21]. To get an optimal value of maximum modularity approximately, simulated annealing was exploited [22], which proceeds by using the partition approach, while the Metropolis sampling used in this paper is applied to estimate the node membership directly.

## 3. Stochastic Block Model and Parameter Estimation

In this paper, we consider a special symmetric stochastic block model (SSBM), which is defined as follows:

**Definition** **1** (SSBM with *k* communities)**.**
*Let 0≤q<p≤1, V=[n] and $X=\left(X_{1},\ldots ,X_{n}\right)\in W^{n}$. X satisfies the constraint that $|\left\{v\in \left[n\right]:X_{v}=u\right\}|=\frac{n}{k}$ for u∈W. The random graph G is generated under SSBM(n,k,p,q) if the following two conditions are satisfied.*


*There is an edge of G between the vertices i and j with probability p if $X_{i}=X_{j}$ and with probability q if $X_{i}\neq X_{j}$.*

*The existences of each edge are mutually independent.*



To explain SSBM in more detail, we define the random variable $Z_{ij}:=1\left[\left\{i,j\right\}\in E\left(G\right)\right]$, which is the indicator function of the existence of an edge between nodes *i* and *j*. Given the node labels *X*, $Z_{ij}$ follows Bernoulli distribution, whose expectation is given by:(1)$$E\left[Z_{ij}\right]=\left\{\begin{array}{cc} p & \mathrm{if}X_{i}=X_{j}\\ q & \mathrm{if}X_{i}\neq X_{j} \end{array}\right.$$

Then the random graph *G* with *n* nodes is completely specified by $Z:=\{Z_{ij},1\leq i<j\leq n\}$ in which all $Z_{ij}$ are jointly independent. The probability distribution for SSBM can be written as:(2)$$\begin{array}{cc} \; & P_{G}\left(G\right):=P_{G}\left(Z=z|X\right)=p^{\sum_{X_{i}=X_{j}}z_{ij}}q^{\sum_{X_{i}\neq X_{j}}z_{ij}}\\ \; & \;\cdot {\left(1-p\right)}^{\sum_{X_{i}=X_{j}}\left(1-z_{ij}\right)}{\left(1-q\right)}^{\sum_{X_{i}\neq X_{j}}\left(1-z_{ij}\right)} \end{array}$$

We will use the notation $G_{n}$ to represent the set containing all graphs with *n* nodes. By the normalization property, $P_{G}\left(G_{n}\right)=\sum_{G\in G_{n}}P_{G}\left(G\right)=1$.

In Definition 1, we have supposed that the node label *X* is fixed instead of a uniformly distributed random variable. Since the maximum posterior estimator is equivalent to the maximum likelihood when the prior is uniform, these two definitions are equivalent. Although the random variable definition is more commonly used in previous literature [6], fixing *X* makes our formal analysis more concise.

Given the SBM, the exact recovery problem can be formally defined as follows:

**Definition** **2** (Exact recovery in SBM)**.**
*Given X, the random graph G is drawn under SSBM(n,k,p,q). We say that the exact recovery is solvable for SSBM(n,k,p,q) if there exists an algorithm that takes G as input and outputs $\hat{X}$ such that:*
$$P_{a}\left(\hat{X}\right):=P\left(\hat{X}\in S_{k}\left(X\right)\right)\to 1\mathrm{as}n\to \infty$$


In the above definition, the notation $P_{a}\left(\hat{X}\right)$ is called the probability of accuracy for estimator $\hat{X}$. Let $P_{e}\left(\hat{X}\right)=1-P_{a}\left(\hat{X}\right)$ represent the probability of error. Definition 2 can also be formulated as $P_{e}\left(\hat{X}\right)\to 0$ as n→∞. The notation $\hat{X}\in S_{k}\left(X\right)$ means that we can only expect a recovery up to a global permutation of the ground truth label vector *X*. This is common in unsupervised learning as no anchor exists to assign labels to different communities. Additionally, given a graph *G*, the algorithm can be either deterministic or stochastic. Generally speaking, the probability of $\hat{X}\in S_{k}\left(X\right)$ should be understood as $\sum_{G\in G_{n}}P_{G}\left(G\right)P_{\hat{X}|G}\left(\hat{X}\in S_{k}\left(X\right)\right)$, which reduced to $P_{G}\left(\hat{X}\in S_{k}\left(X\right)\right)$ for the deterministic algorithm.

For constants p,q, which are irrelevant with the graph size *n*, we can always find algorithms to recover *X* such that the detection error decreases exponentially fast as *n* increases; that is to say, the task with a dense graph is relatively easy to handle. Within this paper, we consider a sparse case when $p=\frac{a\log n}{n},q=\frac{b\log n}{n}$. This case corresponds to the sparsest graph when exact recovery of the SBM is possible. And under this condition, a well known result [9] states that exact recovery is possible if and only if:(3)$$\sqrt{a}-\sqrt{b}>\sqrt{k}$$

Before diving into the exact recovery problem, we first consider the inference problem for SBM. Suppose *k* is known, and we want to estimate a,b from the graph *G*. We offer a simple method by counting the number of edges $T_{1}$ and the number of triangles $T_{2}$ of *G*, and the estimators $\hat{a},\hat{b}$ are obtained by solving the following equation systems: (4)$$\begin{array}{cc} \frac{x+(k-1)y}{2k} & =\frac{T_{1}}{n\log n} \end{array}$$(5)$$\begin{array}{cc} \frac{1}{k^{2}}\left(\frac{x^{3}}{6}+\frac{k-1}{2}xy^{2}+\left(k-1\right)\left(k-2\right)\frac{y^{3}}{6}\right) & =\frac{T_{2}}{\log^{3}n} \end{array}$$

The theoretical guarantee for the solution is given by the following theorem:

**Theorem** **1.**
*When n is large enough, the equation system of Equations (Equation 4) and (5) has the unique solution $\left(\hat{a},\hat{b}\right)$, which are unbiased consistent estimators of (a,b). That is, $E\left[\hat{a}\right]=a,E\left[\hat{b}\right]=b$, and $\hat{a}$ and $\hat{b}$ converge to a,b in probability, respectively.*


Given a graph generated by the SBM, we can use Theorem 1 to obtain the estimated a,b and determine whether exact recovery of label *X* is possible by Equation (Equation 3). Additionally, Theorem 1 provides good estimation of a,b to initialize their parameters of some recovery algorithm like maximum likelihood or our proposed Metropolis sampling in Section 6.

## 4. Ising Model for Community Detection

In the previous section, we have defined SBM and its exact recovery problem. While SBM is regarded as obtaining the graph observation *G* from node label *X*, the Ising model provides a way to generate estimators of *X* from *G* by a stochastic procedure. The definition of such an Ising model is given as follows:

**Definition** **3** (Ising model with *k* states)**.**
*Given a graph G sampled from $\mathrm{SSBM}(n,k,\frac{a\log n}{n},\frac{b\log n}{n})$, the Ising model with parameters γ,β>0 is a probability distribution of the state vector $\sigma \in W^{n}$ whose probability mass function is*
(6)$$\begin{array}{c} P_{\sigma |G}\left(\sigma =\bar{\sigma }\right)=\frac{\exp(-\beta H\left(\bar{\sigma }\right))}{Z_{G}\left(\alpha ,\beta \right)} \end{array}$$
*where*
(7)$$H\left(\bar{\sigma }\right)=\gamma \frac{\log n}{n}\sum_{\left\{i,j\}\notin E(G\right)}\delta \left({\bar{\sigma }}_{i},{\bar{\sigma }}_{j}\right)-\sum_{\left\{i,j\}\in E(G\right)}\delta \left({\bar{\sigma }}_{i},{\bar{\sigma }}_{j}\right)$$

*The subscript in $P_{\sigma |G}$ indicates that the distribution depends on G, and  $Z_{G}\left(\alpha ,\beta \right)$ is the normalizing constant for this distribution.*


In physics, β refers to the inverse temperature and $Z_{G}\left(\gamma ,\beta \right)$ is called the partition function. The Hamiltonian energy $H(\bar{\sigma })$ consists of two terms: the repelling interaction between nodes without edge connection and the attracting interaction between nodes with edge connection. The parameter γ indicates the ratio of the strength of these two interactions. The term $\frac{\log n}{n}$ is added to balance the two interactions because there are only $O(\frac{\log n}{n})$ connecting edges for each node. The probability of each state is proportional to $\exp(-\beta H\left(\bar{\sigma }\right))$, and the state with the largest probability corresponds to that with the lowest energy.

The classical definition of the Ising model is specified by $H\left(\sigma \right)=-\sum_{\left(i,j)\in E(G\right)}\sigma_{i}\cdot \sigma_{j}$ for $\sigma_{i}=\pm 1$. There are two main differences between Definition 3 and the classical one. Firstly, we add a repelling term between nodes without an edge connection. This makes these nodes have a larger probability to take different labels. Secondly, we allow the state at each node to take *k* values from *W* instead of the two values ±1. When γ=0 and k=2, Definition 3 is reduced to the classical definition of the Ising model up to a scaling factor.

Definition 3 gives a stochastic estimator ${\hat{X}}^{*}$ for *X*: ${\hat{X}}^{*}$ is one sample generated from the Ising model, which is denoted as ${\hat{X}}^{*}∼\mathrm{Ising}\left(\gamma ,\beta \right)$. The exact recovery error probability for ${\hat{X}}^{*}$ can be written as $P_{e}\left({\hat{X}}^{*}\right):=\sum_{G\in G_{n}}P_{G}\left(G\right)P_{\sigma |G}\left(S_{k}^{c}\left(X\right)\right)$. From this expression we can see that the error probability is determined by two parameters (γ,β). When these parameters take proper values, $P_{e}\left({\hat{X}}^{*}\right)\to 0$, and the exact recovery of the SBM is achievable. On the contrary, $P_{e}\left({\hat{X}}^{*}\right)\to 1$ if (γ,β) takes other values. These two cases are summarized in the following theorem:

**Theorem** **2.**
*Define the function $g\left(\beta \right),\tilde{g}\left(\beta \right)$ as follows:*
(8)$$g\left(\beta \right)=\frac{be^{\beta }+ae^{-\beta }}{k}-\frac{a+b}{k}+1$$
*and:*
(9)$$\tilde{g}\left(\beta \right)=\left\{\begin{array}{cc} g(\beta ) & \beta \leq \bar{\beta }=\frac{1}{2}\log\frac{a}{b}\\ g\left(\bar{\beta }\right)=1-\frac{{\left(\sqrt{a}-\sqrt{b}\right)}^{2}}{k} & \beta >\bar{\beta } \end{array}\right.$$
*where $\bar{\beta }=\arg\min_{\beta >0}g\left(\beta \right)$. Let $\beta^{*}$ be defined as:*
(10)$$\beta^{*}=\log\left(\frac{a+b-k-\sqrt{{\left(a+b-k\right)}^{2}-4ab)}}{2b}\right)$$
*which is the solution to the equation g(β)=0 and $\beta^{*}<\bar{\beta }$. Then depending on how (γ,β) take values, for any given ϵ>0 and ${\hat{X}}^{*}∼\mathrm{Ising}\left(\gamma ,\beta \right)$, when n is sufficiently large, we have:*


*If γ>b and $\beta >\beta^{*}$, $P_{e}\left({\hat{X}}^{*}\right)\leq n^{\tilde{g}\left(\beta \right)/2+\epsilon }$;*

*If γ>b and $\beta <\beta^{*}$, $P_{a}\left({\hat{X}}^{*}\right)\leq \left(1+o\left(1\right)\right)\max\left\{n^{g(\bar{\beta })},n^{-g(\beta )+\epsilon }\right\}$;*

*If γ<b, $P_{a}\left({\hat{X}}^{*}\right)\leq \exp\left(-Cn\right)$ for any C>0.*



By simple calculus, $\tilde{g}\left(\beta \right)<0$ for $\beta >\beta^{*}$ and g(β)>0 for $\beta <\beta^{*}$. $g(\bar{\beta })<0$ follows from Equation (Equation 3). The illustration of $g\left(\beta \right),\tilde{g}\left(\beta \right)$ is shown in Figure 1a. Therefore, for sufficiently small ϵ and as n→∞, the upper bounds in Theorem 2 all converge to 0 at least in polynomial speed. Therefore, Theorem 2 establishes the sharp phase transition property of the Ising model, which is illustrated in Figure 1b.

Theorem 2 can also be understood from the marginal distribution for $\sigma :P_{\sigma }\left(\sigma =\bar{\sigma }\right)=\sum_{G\in G_{n}}P_{G}\left(G\right)P_{\sigma |G}\left(\sigma =\bar{\sigma }\right)$. Let $D(\sigma ,\sigma^{\prime })$ be the event when σ is closest to $\sigma^{\prime }$ among all its permutations. That is,
(11)$$D\left(\sigma ,\sigma^{\prime }\right):=\left\{\sigma =\arg\min_{f\in S_{k}}\mathrm{dist}\left(f\left(\sigma \right),\sigma^{\prime }\right)\right\}$$

Then Theorem 2 can be stated with respect to the marginal distribution $P_{\sigma }$:

**Corollary** **1.**
*Suppose γ>b, depending on how β takes values:*

*When $\beta >\beta^{*}$, $P_{\sigma }\left(\sigma =X|D\left(\sigma ,X\right)\right)=1-o\left(1\right)$;*

*When $\beta <\beta^{*}$, $P_{\sigma }\left(\sigma =X|D\left(\sigma ,X\right)\right)=o\left(1\right)$.*



Below we outline the proof ideas of Theorem 2. The insight is obtained from the analysis of the one-flip energy difference. This useful result is summarized in the following lemma:

**Lemma** **1.**
*Suppose ${\bar{\sigma }}^{\prime }$ differs from $\bar{\sigma }$ only at position r by ${\bar{\sigma }}_{r}^{\prime }=\omega^{s}\cdot {\bar{\sigma }}_{r}$. Then the change of energy is:*
(12)$$\begin{array}{cc} H\left({\bar{\sigma }}^{\prime }\right)-H\left(\bar{\sigma }\right) & =\left(1+\gamma \frac{\log n}{n}\right)\sum_{i\in N_{r}\left(G\right)}J_{s}\left({\bar{\sigma }}_{r},{\bar{\sigma }}_{i}\right)\\ \; & +\gamma \frac{\log n}{n}\left(m\left(\omega^{s}\cdot {\bar{\sigma }}_{r}\right)-m\left({\bar{\sigma }}_{r}\right)+1\right) \end{array}$$
*where $m\left(\omega^{j}\right):=|\{i\in \left[n\right]|{\bar{\sigma }}_{i}=\omega^{j}\left|\right\}$, $N_{r}\left(G\right):=\left\{j|\left(r,j\right)\in E\left(G\right)\right\}$ and $J_{s}\left(x,y\right)=\delta \left(x,y\right)-\delta \left(\omega_{s}\cdot x,y\right)$.*


Lemma 1 gives an explicit way to compare the probability of two neighboring states by the following equality:(13)$$\frac{P_{\sigma |G}\left(\sigma ={\bar{\sigma }}^{\prime }\right)}{P_{\sigma |G}\left(\sigma =\bar{\sigma }\right)}=\exp\left(-\beta \left(H\left({\bar{\sigma }}^{\prime }\right)-H\left(\bar{\sigma }\right)\right)\right)$$

Additionally, since the graph is sparse and every node has O(logn) neighbors, from Equation (Equation 12) the computational cost (time complexity) for the energy difference is also O(logn).

When $H\left({\bar{\sigma }}^{\prime }\right)>H\left(\bar{\sigma }\right)$, we can expect $P_{\sigma |G}\left(\sigma ={\bar{\sigma }}^{\prime }\right)$ is far less than $P_{\sigma |G}\left(\sigma =\bar{\sigma }\right)$. Roughly speaking, if  $\sum_{\mathrm{dist}(\sigma^{\prime },X)=1}\exp\left(-\beta \left(H\left({\bar{\sigma }}^{\prime }\right)-H\left(X\right)\right)\right)$ converges to zero, we can expect the probability of all other states differing from $S_{k}\left(X\right)$ converges to zero. On the contrary, if $\sum_{\mathrm{dist}(\sigma^{\prime },X)=1}\exp\left(-\beta \left(H\left({\bar{\sigma }}^{\prime }\right)-H\left(X\right)\right)\right)$ tends to infinity, then $P_{\sigma }\left(S_{k}\left(X\right)\right)$ converges to zero. This illustrates the idea behind the proof of Theorem 2. The rigorous proof can be found in Section 8.

## 5. Community Detection via Energy Minimization

Since $\beta^{*}$ is irrelevant with *n*, when γ>b, we can choose a sufficiently large β such that $\beta >\beta^{*}$, then by Theorem 2, $\sigma \in S_{k}\left(X\right)$ almost surely, which implies that $P_{\sigma |G}\left(\sigma =X\right)$ has the largest probability for almost all graphs *G* sampled from the SBM. Therefore, instead of sampling from the Ising model, we can directly maximize the conditional probability to find the state with the largest probability. Equivalently, we can proceed by minimizing the energy term in Equation (Equation 7):(14)$${\hat{X}}^{\prime }:=\arg\min_{\bar{\sigma }\in W^{n}}H\left(\bar{\sigma }\right)$$

In (Equation 14), we allow $\bar{\sigma }$ to take values from $W^{n}$. Since we know *X* has equal size $|\left\{v\in \left[n\right]:X_{v}=u\right\}|=\frac{n}{k}$ for each label *u*, another formulation is to restrict the search space to $W^{*}:=\{\sigma \in W^{n}\left|\right|\left\{v\in \left[n\right]:\sigma_{v}=\omega^{s}\right\}\left|=\frac{n}{k},s=0,\ldots ,k-1\right\}$. When $\sigma \in W^{*}$, minimizing H(σ) is equivalent to:(15)$${\hat{X}}^{\prime \prime }:=\arg\min_{\sigma \in W^{*}}\sum_{\left\{i,j\}\notin E(G\right)}\delta \left(\sigma_{i},\sigma_{j}\right)$$
where the minimal value is the minimum cut between different detected communities.

When ${\hat{X}}^{\prime \prime }\neq X$, we must have $\mathrm{dist}({\hat{X}}^{\prime \prime },X)\geq 2$ to satisfy the constraint ${\hat{X}}^{\prime \prime }\in W^{*}$. Additionally, the estimator of ${\hat{X}}^{\prime \prime }$ is parameter-free whereas ${\hat{X}}^{\prime }$ depends on γ. The extra parameter γ in the expression of ${\hat{X}}^{\prime }$ can be regarded as a kind of Lagrange multiplier for this integer programming problem. Thus, the optimization problem for ${\hat{X}}^{\prime }$ is the relaxation of that for ${\hat{X}}^{\prime \prime }$ by introducing a penalized term and enlarging the searched space from $W^{*}$ to $W^{n}$.

When $\beta >\bar{\beta }$, $\tilde{g}\left(\beta \right)$ becomes a constant value. Therefore, we can get $n^{g(\bar{\beta })/2}$ as the tightest error upper bound for the Ising estimator ${\hat{X}}^{*}$ from Theorem 2. For the estimator ${\hat{X}}^{\prime }$ and ${\hat{X}}^{\prime \prime }$, we can obtain a sharper error upper bound, which is summarized in the following theorem:

**Theorem** **3.**
*When $\sqrt{a}-\sqrt{b}>\sqrt{k}$, for sufficiently large n,*

*If γ>b, $P_{G}\left({\hat{X}}^{\prime }\notin S_{k}\left(X\right)\right)\leq \left(k-1+o\left(1\right)\right)n^{g(\bar{\beta })}$;*

*$P_{G}\left({\hat{X}}^{\prime \prime }\notin S_{k}\left(X\right)\right)\leq \left({\left(k-1\right)}^{2}+o\left(1\right)\right)n^{2g(\bar{\beta })}$.*



As $g(\bar{\beta })<0$, $n^{2g(\bar{\beta })}<n^{g(\bar{\beta })}<n^{g(\bar{\beta })/2}$, Theorem 3 implies that $P_{e}\left({\hat{X}}^{\prime \prime }\right)$ has the sharpest upper bound among the three estimators. This can be intuitively understood as the result of smaller search space. The proof technique of Theorem 3 is to consider the probability of events $H\left(X\right)>H\left(\bar{\sigma }\right)$ for dist(σ,X)≥1. Then by union bound, these error probabilities can be summed up. We note that a loose bound $n^{g(\bar{\beta })/4}$ was obtained in [7] for the estimator ${\hat{X}}^{\prime \prime }$ when k=2. For a general case, since $\tilde{g}\left(\beta \right)=1-\frac{{\left(\sqrt{a}-\sqrt{b}\right)}^{2}}{k}$, Theorem 3 implies that exact recovery is possible using ${\hat{X}}^{\prime }$ as long as $\sqrt{a}-\sqrt{b}>\sqrt{k}$ is satisfied.

Estimator ${\hat{X}}^{\prime }$ has one parameter, γ. When γ takes different values, ${\hat{X}}^{\prime }$ is equivalent with maximum likelihood or maximum modularity in the asymptotic case. The following analysis shows their relationship intuitively.

The maximum likelihood estimator is obtained by maximizing the log-likelihood function. From (Equation 2), this function can be written as:$$\log P_{G}\left(Z|X=\sigma \right)=-\log\frac{a}{b}\cdot H\left(\sigma \right)+C$$
where the parameter γ in H(σ) satisfies $\gamma \frac{\log n}{n}=\frac{1}{\log(a/b)}\left(\log\left(1-\frac{a\log n}{n}\right)-\log\left(1-\frac{b\log n}{n}\right)\right)$ and *C* is a constant irrelevant with σ. When *n* is sufficiently large, we have $\gamma \to \gamma_{ML}:=\frac{a-b}{\log(a/b)}$. That is, the maximum likelihood estimator is equivalent to ${\hat{X}}^{\prime }$ when $\gamma =\gamma_{ML}$ asymptotically.

The maximum modularity estimator is obtained by maximizing the modularity of a graph [23], which is defined by:(16)$$\begin{array}{cc} Q & =\frac{1}{2|E|}\sum_{ij}\left(A_{ij}-\frac{d_{i}d_{j}}{2|E|}\right)\delta \left(C_{i},C_{j}\right) \end{array}$$

For the *i*-th node, $d_{i}$ is its degree and $C_{i}$ is its community belonging. *A* is the adjacency matrix. Up to a scaling factor, the modularity *Q* can be re-written using the label vector σ as:(17)$$\begin{array}{cc} Q(\sigma )= & -\sum_{\left\{i,j\}\notin E(G\right)}\frac{d_{i}d_{j}}{2|E|}\delta \left(\sigma_{i},\sigma_{j}\right)\\ \; & +\sum_{\left\{i,j\}\in E(G\right)}\left(1-\frac{d_{i}d_{j}}{2|E|}\right)\delta \left(\sigma_{i},\sigma_{j}\right) \end{array}$$

From (Equation 17), we can see that Q(σ)→−H(σ) with $\gamma =\gamma_{MQ}=\frac{a+b}{2}$ as n→∞. Indeed, we have $d_{i}∼\frac{(a+b)\log n}{2},\left|E\right|∼\frac{1}{2}nd_{i}$. Therefore, we have $\frac{d_{i}d_{j}}{2|E|}\to \gamma_{MQ}\frac{\log n}{n}$. That is, the maximum modularity estimator is equivalent with ${\hat{X}}^{\prime }$ when $\gamma =\gamma_{MQ}$ asymptotically.

Using a>b and the inequality $x-1>\log x>2\frac{x-1}{x+1}$ for x>1 we can verify that $\gamma_{MQ}>\gamma_{ML}>b$. That is, both the maximum likelihood and the maximum modularity estimator satisfy the exact recovery conditions γ>b in Theorem 3.

## 6. Community Detection Based on Metropolis Sampling

From Theorem 2, if we could sample from the Ising model, then with large probability, the sample is aligned with *X*. However, exact sampling is difficult when *n* is very large since the cardinality of the state space increases in the rate of $k^{n}$. Therefore, some approximation is necessary, and the most common way to generate an Ising sample is using Metropolis sampling [14]. Empirically speaking, starting from a random state, the Metropolis algorithm updates the state by randomly selecting one position to flip its state at each iteration step. Then after some initial burning time, the generated samples can be regarded as sampling from the Ising model.

The theoretical guarantee of Metropolis sampling is based on the Markov chain. Under some general conditions, Metropolis samples converge to the steady state of the Markov chain, and the steady state follows the probability distribution to be approximated. For the Ising model, there are many previous works which have shown the convergence of Metropolis sampling [24].

For our specific Ising model and energy term in Equation (Equation 7), the pseudo code of our algorithm is summarized in Algorithm 1. This algorithm requires that the number of the communities *k* is known and the strength ratio parameter γ is given. We should choose γ>b where *b* is estimated by $\hat{b}$ in Theorem 1. The iteration time *N* should also be specified in advance.
**Algorithm 1** Metropolis sampling algorithm for SBM.Inputs: the graph *G*, inverse temperature β, the strength ratio parameter γOutput: $\hat{X}=\bar{\sigma }$
1:random initialize $\bar{\sigma }\in W^{n}$2:**for**i=1,2,…,N**do**3: propose a new state ${\bar{\sigma }}^{\prime }$ according to Lemma 1 where s,r are randomly chosen4: compute $\Delta H\left(r,s\right)=H\left({\bar{\sigma }}^{\prime }\right)-H\left(\bar{\sigma }\right)$ using (Equation 12)5: **if**
ΔH(r,s)<0
**then**
6:  $\sigma_{r}\leftarrow w^{s}\cdot \sigma_{r}$
7: **else**
8:  with probability exp(−βΔH(r,s)) such that $\sigma_{r}\leftarrow w^{s}\cdot \sigma_{r}$9: **end if**
10:**end for**

The computation of ΔH(r,s) needs O(logn) time from Lemma 1. For some special Ising model, it needs to take N=O(nlogn) to generate the sample for good approximation [25]. For our model, it is unknown whether O(nlogn) is sufficient, and we empirically chose $N=O(n^{2})$ in numerical experiments. Then the time complexity of Algorithm 1 is $O(n^{2}\log n)$.

In the remaining part of this section, we present experiments conducted to verify our theoretical results. Firstly, we considered several combinations of (a,b,k) and obtained the estimator $\left(\hat{a},\hat{b}\right)$ by Theorem 1. Using the empirical mean squared error (MSE) $\frac{1}{m}\sum_{i=1}^{m}{\left(\hat{a}-a\right)}^{2}+{\left(\hat{b}-b\right)}^{2}$ as the criterion and choosing m=1000, the result is shown in Figure 2a. As we can see, as *n* increases, the MSE decreases polynomially fast. Therefore, the convergence of $\hat{a}\to a$ and $\hat{b}\to b$ was verified.

Secondly, using Metropolis sampling, we conducted a moderate simulation to verify Theorem 2 for the case γ>b. We chose n=9000,k=2, and the empirical accuracy was computed by $P_{e}=\frac{1}{m_{1}m_{2}}\sum_{i=1}^{m_{1}}\sum_{j=1}^{m_{2}}1\left[{\hat{X}}^{*}=\pm X\right]$. In this formula, $m_{1}$ is the number of times the random graph was generated by the SBM, whereas $m_{2}$ is the number of times consecutive samples were generated by Algorithm 1 for a given graph. We chose $m_{1}=2100,m_{2}=6000$, which is fairly large and can achieve a good approximation of $P_{e}\left({\hat{X}}^{*}\right)$ by the law of large numbers. The result is shown in Figure 2b.

The vertical red line ($\beta =\beta^{*}=0.198$), computed from (Equation 10), represents the phase transition threshold. The point $\left(0.199,\frac{1}{2}\right)$ in the figure can be regarded as the empirical phase transition threshold, whose first coordinate is close to $\beta^{*}$. The green line $\left(\beta ,n^{g(\beta )/2}\right)$ is the theoretical lower bound of accuracy for $\beta >\beta^{*}$, and the purple line $\left(\beta ,n^{-g(\beta )}\right)$ is the theoretical upper bound of accuracy for $\beta <\beta^{*}$. It can be expected that as *n* becomes larger, the empirical accuracy curve (blue line in the figure) will approach the step function, which jumps from 0 to 1 at $\beta =\beta^{*}$.

## 7. Conclusions

In this paper, we presented one convergent estimator (in Theorem 1) to infer the parameters of the SBM and analyzed three label estimators to detect communities of the SBM. We gave the exact recovery error upper bound for all label estimators (in Theorems 2 and 3) and studied their relationships. By introducing the Ising model, our work makes a new path to study the exact recovery problem for the SBM. More theoretical and empirical work will be done in the future, such as convergence analyses on modularity (in Equation (Equation 17)), the necessary iteration time (in Algorithm 1) for Metropolis sampling, and so on.

## 8. Proof of Main Theorems

### 8.1. Proof of Theorem 1

**Lemma** **2.**
*Consider an Erdős–Rényi random graph G with n nodes, in which edges are placed independently with probability p [26]. Suppose $p=\frac{a\log n}{n}$, the number of edges is denoted by |E| while the number of triangles is T. Then $\frac{\left|E\right|}{n\log n}\to \frac{a}{2}$ and $\frac{T}{\log^{3}n}\to \frac{a^{3}}{6}$ in probability.*


**Proof.** Let $X_{ij}$ represent a Bernoulli random variable with parameter *p*. Then $\left|E\right|=\sum_{i,j}X_{ij}$, $X_{ij}$ are i.i.d. $E\left[T\left(G\right)\right]=\frac{n(n-1)}{2}p=\frac{(n-1)\log n}{2}a$ and $\mathrm{Var}\left[|E|\right]=\frac{n(n-1)}{2}p\left(1-p\right)<a\frac{(n-1)\log n}{2}$. Then by Chebyshev’s inequality,
$$\begin{array}{cc} P(|\frac{\left|E\right|}{n\log n}-\frac{a}{2}\frac{n-1}{n}|>\epsilon ) & \leq \frac{\mathrm{Var}[|E|/(n\log n)]}{\epsilon^{2}}\\ \; & <\frac{a(n-1)}{2n^{2}\epsilon^{2}\log n} \end{array}$$For a given ϵ, when *n* is sufficiently large,
$$\begin{array}{cc} P(|\frac{\left|E\right|}{n\log n}-\frac{a}{2}|>\epsilon ) & <P(|\frac{\left|E\right|}{n\log n}-\frac{a}{2}\frac{n-1}{n}|>2\epsilon )\\ \; & \leq \frac{n-1}{8n^{2}\epsilon^{2}\log n} \end{array}$$Therefore, by the definition of convergence in probability, we have $\frac{\left|E\right|}{n\log n}\to \frac{a}{2}$ as n→∞.Let $X_{ijk}$ represents a Bernoulli random variable with parameter $p^{3}$. Then $T=\sum_{i,j,k}X_{ijk}$. It is easy to compute that E[T]=n3p3. Since $X_{ijk}$ are not independent, the variance of *T* needs careful calculation. From [27] we know that:
Var[T]=n3p3+12n4p5+30n5p6+20n6p6−n32p6=O(log3n)Therefore by Chebyshev’s inequality,
$$\begin{array}{cc} P(|\frac{T}{\log^{3}n}-\frac{a^{3}}{6}\frac{\left(n-1)(n-2\right)}{n^{2}}|>\epsilon ) & \leq \frac{\mathrm{Var}[T/\log^{3}n]}{\epsilon^{2}}\\ \; & =\frac{1}{\epsilon^{2}}O\left(\frac{1}{\log^{3}n}\right) \end{array}$$Hence, $\frac{T}{\log^{3}n}\to \frac{a^{3}}{6}$. □

The convergence of |E| in the Erdős–Rényi graph can be extended directly to the SBM since the existence of each edge is independent. However, for *T*, it is a little tricky since the existences of each triangle are mutually dependent. The following two lemmas give the formula for the variance of inter-community triangles in the SBM.

**Lemma** **3.**
*Consider a two-community SBM (2n,p,q) and count the number of triangles T, which has a node in $S_{1}$ and an edge in $S_{2}$. Then the variance of T is:*
(18)$$\begin{array}{cc} \mathrm{Var}[T] & =\frac{n^{2}\left(n-1\right)}{2}q^{2}p+n^{2}\left(n-1\right)\left(n-2\right)p^{2}q^{3}\\ \; & +\frac{n^{2}{\left(n-1\right)}^{2}}{2}q^{4}p-\frac{n^{2}\left(n-1\right)\left(3n-4\right)}{2}q^{4}p^{2} \end{array}$$


**Lemma** **4.**
*Consider a three-community SBM (3n,p,q) and count the number of triangles T, which has a node in $S_{1}$, one node in $S_{2}$, and one node in $S_{3}$. Then the variance of T is:*
$$\mathrm{Var}\left[T\right]=n^{3}q^{3}+3n^{3}\left(n-1\right)q^{4}+3n^{3}{\left(n-1\right)}^{2}q^{5}-n^{3}\left(3n^{2}-3n+1\right)q^{6}$$


The proof of the above two lemmas uses some counting techniques and is similar to that in [27], and we omit it here.

**Lemma** **5.**
*For a SBM (n,k,p,q) where $p=\frac{a\log n}{n},q=\frac{b\log n}{n}$. The number of triangles is T. Then $\frac{T}{{\left(\log n\right)}^{3}}$ converges to $\frac{1}{k^{2}}\left(\frac{a^{3}}{6}+\frac{k-1}{2}ab^{2}+\left(k-1\right)\left(k-2\right)\frac{b^{3}}{6}\right)$ in probability as n→∞.*


**Proof.** We split *T* into three parts: the first is the number of triangles within community *i*, $T_{i}$. There are *k* terms of $T_{i}$. The second is the number of triangles that have one node in community *i* and one edge in community *j*, $T_{ij}$. There are k(k−1) terms of $T_{ij}$. The third is the number of triangles that have one node in community *i*, one node in community *j* and one node in community *k*.We only need to show that:
(19)$$\begin{array}{cc} \frac{T_{i}}{\log^{3}n} & \to \frac{{\left(a/k\right)}^{3}}{6} \end{array}$$
(20)$$\begin{array}{cc} \frac{T_{ij}}{\log^{3}n} & \to \frac{1}{2}\left(a/k\right){\left(b/k\right)}^{2} \end{array}$$
(21)$$\begin{array}{cc} \frac{T_{ijk}}{\log^{3}n} & \to {\left(b/k\right)}^{3} \end{array}$$The convergence of $\frac{T_{i}}{\log^{3}n}$ comes from Lemma 2. For $T_{ij}$ we use the conclusion from Lemma 3. We replace *n* with n/k, $p=a\frac{\log n}{n}$, and $q=b\frac{\log n}{n}$ in Equation (Equation 18). $\mathrm{Var}\left[T_{ij}\right]∼\frac{ab^{2}}{2k^{3}}\log^{3}n$. Since the expectation of $\frac{T_{ij}}{\log^{3}n}$ is (n/k)n/k2pq2/(log3n)=n−12nab2k3, by Chebyshev’s inequality we can show that:
$$\begin{array}{cc} P(|\frac{T_{ij}}{\log^{3}n}-\frac{n-1}{2n}\frac{ab^{2}}{k^{3}}|>\epsilon ) & \leq \frac{\mathrm{Var}[T_{ij}/\log^{3}n]}{\epsilon^{2}}\\ \; & =\frac{1}{\epsilon^{2}}O\left(\frac{1}{\log^{3}n}\right) \end{array}$$Therefore, $\frac{T_{ij}}{\log^{3}n}$ converges to $\frac{1}{2}\left(a/k\right){\left(b/k\right)}^{2}$.To prove $\frac{T_{ijk}}{\log^{3}n}\to {\left(b/k\right)}^{3}$, from Lemma 4 we can get $\mathrm{Var}\left[T_{ijk}\right]=O\left(\log^{5}n\right)$:
$$P\left(|\frac{T_{ijk}}{\log^{3}n}-\frac{b^{3}}{k^{3}}|>\epsilon \right)\leq \frac{\mathrm{Var}[T_{ijk}/\log^{3}n]}{\epsilon^{2}}=\frac{1}{\epsilon^{2}}O\left(\frac{1}{\log n}\right)$$□

**Proof of Theorem** **1.**Let $e_{1}^{*}=\frac{a+(k-1)b}{2k}$, $k^{2}e_{2}^{*}=\frac{a^{3}}{6}+\frac{k-1}{2}ab^{2}+\left(k-1\right)\left(k-2\right)\frac{b^{3}}{6}$ and $e_{1}=\frac{T_{1}}{n\log n},e_{2}=\frac{T_{2}}{\log^{3}n}$. From Lemma 2, $e_{1}\to e_{1}^{*}$. From Lemma 5, $e_{2}\to e_{2}^{*}$ as n→∞. Using $x=2ke_{1}-\left(k-1\right)y$, we can get:
(22)$$g\left(y\right):=\left(k-1\right)\left(y^{3}-6e_{1}y^{2}+12e_{1}^{2}y\right)+6e_{2}-8ke_{1}^{3}=0$$This equation has a unique real root since g(y) is increasing on R: $g^{\prime }\left(y\right)=3\left(k-1\right){\left(y-2e_{1}\right)}^{2}\geq 0$. Next we show that the root lies within $\left(0,2e_{1}\right)$.
$$\begin{array}{cc} \; & \lim_{n\to \infty }g\left(0\right)=6e_{2}^{*}-8k{\left(e_{1}^{*}\right)}^{3}=-\frac{3}{k^{2}}\left(k-1\right)\left(k-2\right)ab^{2}\\ \; & -\frac{3(k-1)}{k^{2}}a^{2}b-\frac{k-1}{k^{2}}\left(\left(k-2\right)-{\left(k-1\right)}^{2}\right)b^{3}<0\\ \; & \lim_{n\to \infty }g\left(2e_{1}\right)=6e_{2}^{*}-8{\left(e_{1}^{*}\right)}^{3}=\frac{\left(k-1\right){\left(a-b\right)}^{3}}{k^{3}}>0 \end{array}$$Therefore, we can get a unique solution *y* within $\left(0,2e_{1}\right)$. Since (a,b) is a solution for the equation array, the conclusion follows.By taking expectation on both sizes of Equations (Equation 4) and (5) we can show $E\left[\hat{a}\right]=a,E\left[\hat{b}\right]=b$. By the continuous property of g(y), $\hat{b}\to b$ and $\hat{a}\to a$ follows similarly. □

### 8.2. Proof of Theorem 2

**Proof of Lemma** **1.**First we rewrite the energy term in (Equation 7) as:
$$H\left(\bar{\sigma }\right)=\gamma \frac{\log n}{n}\sum_{i<j}\delta \left({\bar{\sigma }}_{i},{\bar{\sigma }}_{j}\right)-\left(1+\gamma \frac{\log n}{n}\right)\sum_{\left\{i,j\}\in E(G\right)}\delta \left({\bar{\sigma }}_{i},{\bar{\sigma }}_{j}\right)$$Then calculating the energy difference term by:
$$\begin{array}{cc} H\left({\bar{\sigma }}^{\prime }\right)-H\left(\bar{\sigma }\right) & =(1+\gamma \frac{\log n}{n})\\ \; & \cdot \sum_{i\in N_{r}\left(G\right)}\left(\delta \left({\bar{\sigma }}_{r},{\bar{\sigma }}_{i}\right)-\delta \left(\omega^{s}\cdot {\bar{\sigma }}_{r},{\bar{\sigma }}_{i}\right)\right)\\ \; & +\gamma \frac{\log n}{n}\sum_{i\neq r}\left(\delta \left(\omega^{s}\cdot {\bar{\sigma }}_{r},{\bar{\sigma }}_{i}\right)-\delta \left({\bar{\sigma }}_{r},{\bar{\sigma }}_{i}\right)\right)\\ \; & =\left(1+\gamma \frac{\log n}{n}\right)\sum_{i\in N_{r}\left(G\right)}J_{s}\left({\bar{\sigma }}_{r},{\bar{\sigma }}_{i}\right)\\ \; & +\gamma \frac{\log n}{n}\sum_{i=1}^{n}\left(\delta \left(\omega^{s}\cdot {\bar{\sigma }}_{r},{\bar{\sigma }}_{i}\right)-\delta \left({\bar{\sigma }}_{r},{\bar{\sigma }}_{i}\right)+1\right)\\ \; & =\left(1+\gamma \frac{\log n}{n}\right)\sum_{i\in N_{r}\left(G\right)}J_{s}\left({\bar{\sigma }}_{r},{\bar{\sigma }}_{i}\right)\\ \; & +\gamma \frac{\log n}{n}\left(m\left(\omega^{s}\cdot {\bar{\sigma }}_{r}\right)-m\left({\bar{\sigma }}_{r}\right)+1\right) \end{array}$$□

Before diving into the technical proof of Theorem 2, we need to introduce some extra notations. When $\bar{\sigma }$ differs from *X* only at position *r*, taking $\bar{\sigma }=X$ in Lemma 1, we have:(23)$$H\left({\bar{\sigma }}^{\prime }\right)-H\left(\bar{\sigma }\right)=\left(1+\gamma \frac{\log n}{n}\right)\left(A_{r}^{0}-A_{r}^{s}\right)+\gamma \frac{\log n}{n}$$
where $A_{r}^{s}$ is defined as $A_{r}^{s}=\left|\left\{j\in \left[n\right]\backslash \left\{r\right\}:\left\{j,r\right\}\in E\left(G\right),X_{j}=\omega^{s}\cdot X_{r}\right\}\right|$. Since the existence of each edge in *G* is independent, $A_{r}^{s}∼\mathrm{Bernoulli}\left(\frac{n}{k},\frac{b\log n}{n}\right)$ for s≠0 and $A_{0}^{s}∼\mathrm{Bernoulli}\left(\frac{n}{k}-1,\frac{a\log n}{n}\right)$.

For the general case, we can write:(24)$$H\left(\bar{\sigma }\right)-H\left(X\right)=\left(1+\gamma \frac{\log n}{n}\right)\left[A_{\bar{\sigma }}-B_{\bar{\sigma }}\right]+\gamma \frac{\log n}{n}N_{\bar{\sigma }}$$
in which we use $A_{\bar{\sigma }}$ or $B_{\bar{\sigma }}$ to represent the binomial random variable with parameter $\frac{a\log n}{n}$ or $\frac{b\log n}{n}$, respectively, and $N_{\bar{\sigma }}$ is a deterministic positive number depending on $\bar{\sigma }$ but irrelevant with the graph structure. The following lemma gives the expression of $A_{\bar{\sigma }},B_{\bar{\sigma }}$ and $N_{\bar{\sigma }}$:

**Lemma** **6.**
*For SSBM (n,k,p,q), we assume $\bar{\sigma }$ differs from the ground truth label vector X in the $\left|I|:=\mathrm{dist}(\bar{\sigma },X\right)$ coordinate. Let $I_{ij}=|\left\{r\in \left[n\right]|X_{r}=w^{i},\sigma_{r}=w^{j}\right\}$ for i≠j and $I_{ii}=0$. We further denote the row sum as $I_{i}=\sum_{j=0}^{k-1}I_{ij}$ and the column sum as $I_{i}^{\prime }=\sum_{j=0}^{k-1}I_{ji}$. Then:*
(25)$$\begin{array}{cc} N_{\bar{\sigma }} & =\frac{1}{2}\sum_{i=0}^{k-1}{\left(I_{i}-I_{i}^{\prime }\right)}^{2} \end{array}$$
(26)$$\begin{array}{cc} B_{\bar{\sigma }} & ∼Bernoulli(\frac{n}{k}|I|+\frac{1}{2}\sum_{i=0}^{k-1}\left(-2I_{i}^{\prime }I_{i}+I_{i}^{\prime 2}-\sum_{j=0}^{k-1}I_{ji}^{2}\right),q) \end{array}$$
(27)$$\begin{array}{cc} A_{\bar{\sigma }} & ∼Bernoulli(\frac{n}{k}|I|-\frac{1}{2}\sum_{i=0}^{k-1}\left(I_{i}^{2}+\sum_{j=0}^{k-1}I_{ij}^{2}\right),p) \end{array}$$


The proof of Lemma 6 is mainly composed of careful counting techniques, and we omit it here. When |I| is small compared to *n*, we have the following Lemma, which is an extension of Proposition 6 in [12].

**Lemma** **7.**
*For $t\in [\frac{1}{k}\left(b-a\right),0]$ and $\left|I\right|\leq n/\sqrt{\log n}$*
(28)$$\begin{array}{cc} \; & P_{G}(B_{\bar{\sigma }}-A_{\bar{\sigma }}\geq t|I|\log n)\\ \leq & \exp\left(|I|\log n\left(f_{\beta }\left(t\right)-\beta t-1+O\left(\frac{1}{\sqrt{\log n}}\right)\right)\right) \end{array}$$
*where $f_{\beta }\left(t\right)=\min_{s\geq 0}\left(g\left(s\right)-st\right)+\beta t\leq \tilde{g}\left(\beta \right)$.*


Corresponding to the three cases of Theorem 2, we use three non-trivial lemmas to establish the properties of the Ising model.

**Lemma** **8.**
*Let γ>b. When $\mathrm{dist}\left(\bar{\sigma },X\right)\geq \frac{n}{\sqrt{\log n}}$ and $D(\bar{\sigma },X)$, the event $P_{\sigma |G}\left(\sigma =\bar{\sigma }\right)>\exp\left(-Cn\right)P_{\sigma |G}\left(\sigma =X\right)$ happens with a probability (with respect to SSBM) less than $\exp(-\tau \left(\alpha ,\beta \right)n\sqrt{\log n})$, where C is an arbitrary constant and τ(α,β) is a positive number.*


**Proof.** We denote the event $P_{\sigma |G}\left(\sigma =\bar{\sigma }\right)>\exp\left(-Cn\right)P_{\sigma |G}\left(\sigma =X\right)$ as $\tilde{D}\left(\bar{\sigma },C\right)$. By Equation (Equation 24), $\tilde{D}\left(\bar{\sigma },C\right)$ is equivalent to:
(29)$$\left(1+\frac{\gamma \log n}{n}\right)\left[B_{\bar{\sigma }}-A_{\bar{\sigma }}\right]>\frac{\gamma \log n}{n}N_{\bar{\sigma }}-\frac{C}{\beta }n$$We claim that $\bar{\sigma }$ must satisfy at least one of the following two conditions:
∃i≠j s.t. $\frac{1}{k(k-1)}\frac{n}{\sqrt{\log n}}\leq I_{ij}\leq \frac{n}{k}-\frac{1}{k(k-1)}\frac{n}{\sqrt{\log n}}$∃i≠j s.t. $I_{ij}>\frac{n}{k}-\frac{1}{k(k-1)}\frac{n}{\sqrt{\log n}}$ and $I_{ji}<\frac{1}{k(k-1)}\frac{n}{\sqrt{\log n}}$If neither of the above two condition holds, then from condition 1 we have $I_{ij}<\frac{1}{k(k-1)}\frac{n}{\sqrt{\log n}}$ or $I_{ij}>\frac{n}{k}-\frac{1}{k(k-1)}\frac{n}{\sqrt{\log n}}$ for any 0≤i,j≤k−1. Since $\sum_{i,j}I_{ij}=\left|I\right|\geq \frac{n}{\sqrt{\log n}}$, there exists i,j such that $I_{ij}>\frac{n}{k}-\frac{1}{k(k-1)}\frac{n}{\sqrt{\log n}}$. Under such conditions, we also assume $I_{ji}>\frac{n}{k}-\frac{1}{k(k-1)}\frac{n}{\sqrt{\log n}}$. Let $X^{\prime }$ be the vector that exchanges the value of $w^{i}$ with $w^{j}$ in *X*. We consider:
(30)$$\begin{array}{cc} \mathrm{dist}(\bar{\sigma },X^{\prime }) & -\mathrm{dist}\left(\bar{\sigma },X\right)=\left|\left\{r\in \left[n\right]|X_{r}=w^{i},{\bar{\sigma }}_{r}\neq w^{j}\right\}\right|\\ \; & +|\{r\in \left[n\right]|X_{r}=w^{j},{\bar{\sigma }}_{r}\neq w^{i}\}|\\ \; & -|\{r\in \left[n\right]|X_{r}=w^{i},{\bar{\sigma }}_{r}\neq w^{i}\}|\\ \; & -|\{r\in \left[n\right]|X_{r}=w^{j},{\bar{\sigma }}_{r}\neq w^{j}\}|\\ \; & =\frac{n}{k}-I_{ij}+\frac{n}{k}-I_{ji}-I_{i}-I_{j}\\ \; & <\frac{2}{k(k-1)}\frac{n}{\sqrt{\log n}}-I_{i}-I_{j}<0 \end{array}$$
which contracts with the fact that $\bar{\sigma }$ is nearest to *X*. Therefore, we should have $I_{ji}<\frac{1}{k(k-1)}\frac{n}{\sqrt{\log n}}$. Now the (i,j) pair satisfies condition 2, which contracts with the fact that $\bar{\sigma }$ satisfies neither of the two conditions.Under condition 1, we can get a lower bound on $|A_{\bar{\sigma }}|$ from Equation (27). Let $I_{ij}^{\prime }=I_{ij}$ for i≠j and $I_{ii}^{\prime }=\frac{n}{k}-I_{i}$. Then we can simplify $|A_{\bar{\sigma }}|$ as:
$$\begin{array}{cc} |A_{\bar{\sigma }}| & =\frac{n}{k}\left|I\right|-\frac{1}{2}\sum_{i=0}^{k-1}\left(I_{i}^{2}+\sum_{j=0}^{k-1}I_{ij}^{2}\right)\\ \; & =\frac{n^{2}}{2k}-\frac{1}{2}\sum_{i=0}^{k-1}\sum_{j=0}^{k-1}I_{ij}^{\prime 2} \end{array}$$We further have $\sum_{i=0}^{k-1}\sum_{j=0}^{k-1}{I^{\prime }}_{ij}^{2}\leq \left(k-1\right)\frac{n^{2}}{k^{2}}+{\left(\frac{n}{k}-I_{ij}\right)}^{2}+I_{ij}^{2}$ where $I_{ij}$ satisfies condition 1. Therefore, $\sum_{i=0}^{k-1}\sum_{j=0}^{k-1}{I^{\prime }}_{ij}^{2}\leq \left(k-1\right)\frac{n^{2}}{k^{2}}+{\left(\frac{1}{k(k-1)}\frac{n}{\sqrt{\log n}}\right)}^{2}+{\left(\frac{n}{k}-\frac{1}{k(k-1)}\frac{n}{\sqrt{\log n}}\right)}^{2}=\frac{n^{2}}{k}-\frac{2n^{2}}{k^{2}\left(k-1\right)\sqrt{\log n}}\left(1+o\left(1\right)\right)$. As a result,
(31)$$A_{\bar{\sigma }}\geq \frac{n^{2}}{k^{2}\left(k-1\right)\sqrt{\log n}}\left(1+o\left(1\right)\right)$$Under condition 2, we can get a lower bound on $N_{\bar{\sigma }}$. Since $\mathrm{dist}\left(\bar{\sigma },X^{\prime }\right)-\mathrm{dist}\left(\bar{\sigma },X\right)\geq 0$, from (Equation 30) we have $I_{ij}+I_{ji}+I_{i}+I_{j}\leq \frac{2n}{k}$. Since $I_{i}\geq I_{ij}>\frac{n}{k}-\frac{1}{k(k-1)}\frac{n}{\sqrt{\log n}}$, we have $I_{j}\leq \frac{2}{k(k-1)}\frac{n}{\sqrt{\log n}}$. Now consider $I_{j}^{\prime }-I_{j}\geq \frac{n}{k}-\frac{3}{k(k-1)}\frac{n}{\sqrt{\log n}}$. From (Equation 25): $N_{\bar{\sigma }}\geq \frac{1}{2}{\left(\frac{n}{k}-\frac{3}{k(k-1)}\frac{n}{\sqrt{\log n}}\right)}^{2}=\frac{n^{2}}{2k^{2}}\left(1+o\left(1\right)\right)$.Now we use the Chernoff inequality to bound Equation (Equation 29); we can omit $\frac{\gamma \log n}{n}$ on the left-hand side since it is far smaller than 1. Let $Z∼\mathrm{Bernoulli}\left(\frac{a\log n}{n}\right),Z^{\prime }∼\mathrm{Bernoulli}\left(\frac{b\log n}{n}\right)$, then:
$$\begin{array}{cc} \; & P_{G}\left(\tilde{D}\left(\bar{\sigma },C\right)\right)\leq {\left(E\left[\exp\left(sZ\right)\right]\right)}^{|B_{\bar{\sigma }}|}{\left(E\left[\exp\left(-sZ^{\prime }\right)\right]\right)}^{|A_{\bar{\sigma }}|}\\ \; & \cdot \exp(-s\left(\frac{\gamma \log n}{n}N_{\bar{\sigma }}-\frac{C}{\beta }n\right))\\ \; & \leq \exp(|B_{\bar{\sigma }}|\frac{b\log n}{n}\left(e^{s}-1\right)+\left|A_{\bar{\sigma }}\right|\frac{a\log n}{n}\left(e^{-s}-1\right)\\ \; & -s(\frac{\gamma \log n}{n}N_{\bar{\sigma }}-\frac{C}{\beta }n)) \end{array}$$Using $|B_{\bar{\sigma }}|=N_{\bar{\sigma }}+\left|A_{\bar{\sigma }}\right|$ we can further simplify the exponential term as:
$$\frac{\log n}{n}\left[\right|A_{\bar{\sigma }}\left|\left(b\left(e^{s}-1\right)+a\left(e^{-s}-1\right)\right)+N_{\bar{\sigma }}\left(b\left(e^{s}-1\right)-\gamma s\right)\right]+s\frac{C}{\beta }n$$Now we investigate the function $g_{1}\left(s\right)=b\left(e^{s}-1\right)+a\left(e^{-s}-1\right)$ and $g_{2}\left(s\right)=b\left(e^{s}-1\right)-\gamma s$. Both functions take zero values at s=0 and $g_{1}^{\prime }\left(s\right)=\left(be^{s}-ae^{-s}\right),g_{2}^{\prime }\left(s\right)=be^{s}-\gamma$. Therefore, $g_{1}^{\prime }\left(0\right)=b-a<0,g_{2}^{\prime }\left(0\right)=b-\gamma <0$ and we can choose $s^{*}>0$ such that $g_{1}\left(s^{*}\right)<0,g_{2}\left(s^{*}\right)<0$. To compensate the influence of the term sCn/β we only need to make sure that the order of $\frac{\log n}{n}\min\{|A_{\bar{\sigma }}\left|,N_{\bar{\sigma }}\right\}$ is larger than *n*. This requirement is satisfied since either $|A_{\bar{\sigma }}|\geq \frac{n^{2}}{k^{2}\left(k-1\right)\sqrt{\log n}}\left(1+o\left(1\right)\right)$ or $N_{\bar{\sigma }}\geq \frac{n^{2}}{2k^{2}}\left(1+o\left(1\right)\right)$. □

**Lemma** **9.**
*If γ>b, $\beta >\beta^{*}$, For $1\leq r\leq \frac{n}{\sqrt{\log n}}$ and ∀ϵ>0, there is a set $G^{\left(r\right)}$ such that:*
(32)$$P_{G}\left(G_{n}^{\left(r\right)}\right)\geq 1-n^{r(\tilde{g}\left(\beta \right)/2+\epsilon )}$$
*and for every $G\in G_{n}^{\left(r\right)}$,*
(33)$$\frac{P_{\sigma |G}\left(\mathrm{dist}\left(\sigma ,X\right)=r|D\left(\sigma ,X\right)\right)}{P_{\sigma |G}\left(\sigma =X|D\left(\sigma ,X\right)\right)}<n^{r\tilde{g}\left(\beta \right)/2}$$

*For $r>\frac{n}{\sqrt{\log n}}$, there is a set $G^{\left(r\right)}$ such that:*
(34)$$P\left(G\in G_{n}^{\left(r\right)}\right)\geq 1-e^{-n}$$
*and for every $G\in G_{n}^{\left(r\right)}$,*
(35)$$\frac{P_{\sigma |G}\left(\mathrm{dist}\left(\sigma ,X\right)=r|D\left(\sigma ,X\right)\right)}{P_{\sigma |G}\left(\sigma =X|D\left(\sigma ,X\right)\right)}<e^{-n}$$


**Proof.** We distinguish the discussion between two cases: $r\leq \frac{n}{\sqrt{\log n}}$ and $r>\frac{n}{\sqrt{\log n}}$.When $r\leq \frac{n}{\sqrt{\log n}}$, we can show that dist(σ,X)=r implies D(σ,X) by using the triangle inequality of dist. For $f\in S_{k}\backslash \left\{\mathrm{id}\right\}$, where id is the identity mapping, we have:
$$\frac{2n}{k}\leq \mathrm{dist}\left(f\left(X\right),X\right)\leq \mathrm{dist}\left(\sigma ,f\left(X\right)\right)+\mathrm{dist}\left(\sigma ,X\right)$$Therefore, $\mathrm{dist}\left(\sigma ,f\left(X\right)\right)\geq \frac{2n}{k}-\frac{n}{\sqrt{\log n}}\geq \mathrm{dist}\left(\sigma ,X\right)$ and Equation (Equation 33) is equivalent with:
(36)$$\frac{P_{\sigma |G}\left(\mathrm{dist}\left(\sigma ,X\right)=r\right)}{P_{\sigma |G}\left(\sigma =X\right)}<n^{r\tilde{g}\left(\beta \right)/2}$$The left-hand side can be written as:
$$\begin{array}{cc} \frac{P_{\sigma |G}\left(\mathrm{dist}\left(\sigma ,X\right)=r\right)}{P_{\sigma |G}\left(\sigma =X\right)} & =\sum_{\mathrm{dist}(\bar{\sigma },X)=r}\exp\left(-\beta \left(H\left(\bar{\sigma }\right)-H\left(X\right)\right)\right)\\ \mathrm{by}(24) & \leq \sum_{\mathrm{dist}(\bar{\sigma },X)=r}\exp\left(\beta_{n}\left(B_{\bar{\sigma }}-A_{\bar{\sigma }}\right)\right) \end{array}$$
where $\beta_{n}=\beta \left(1+\gamma \frac{\log n}{n}\right)$.Define $\Xi_{n}\left(r\right):=\sum_{\mathrm{dist}(\bar{\sigma },X)=r}\exp\left(\beta_{n}\left(B_{\bar{\sigma }}-A_{\bar{\sigma }}\right)\right)$ and we only need to show that:
(37)$$P_{G}\left(\Xi_{n}\left(r\right)\geq n^{r\tilde{g}\left(\beta \right)/2}\right)\leq n^{r(\tilde{g}\left(\beta \right)/2+\epsilon )}$$Define the event $\Lambda_{n}\left(G,r\right):=\left\{B_{\bar{\sigma }}-A_{\bar{\sigma }}<0,\forall \bar{\sigma }\;s.t.\mathrm{dist}\left(\bar{\sigma },X\right)=r\right\}$, and we proceed as follows:
$$\begin{array}{cc} P_{G}\left(\Xi_{n}\left(r\right)\geq n^{r\tilde{g}\left(\beta \right)/2}\right) & \leq P_{G}\left(\Lambda_{n}{\left(G,r\right)}^{c}\right)\\ \; & +P_{G}\left(\Xi_{n}\left(r\right)\geq n^{r\tilde{g}\left(\beta \right)/2}|\Lambda_{n}\left(G,r\right)\right) \end{array}$$For the first term, since $|\left\{\bar{\sigma }|\mathrm{dist}\left(\bar{\sigma },X\right)=r\right\}|={\left(k-1\right)}^{r}n^{r}$, by Lemma 7, $P_{G}\left(\Lambda_{n}{\left(G,r\right)}^{c}\right)\leq {\left(k-1\right)}^{r}n^{rg(\bar{\beta })}\leq n^{r(\tilde{g}\left(\beta \right)/2+\epsilon /2)}$. For the second term, we use Markov inequality:
$$\begin{array}{c} P_{G}\left(\Xi_{n}\left(r\right)\geq n^{r\tilde{g}\left(\beta \right)/2}|\Lambda_{n}\left(G,r\right)\right)\leq E\left[\Xi_{n}\left(r\right)|\Lambda_{n}\left(G,r\right)\right]n^{-r\tilde{g}\left(\beta \right)/2} \end{array}$$The conditional expectation can be estimated as follows:
$$\begin{array}{cc} \; & E\left[\Xi_{n}\left(r\right)|\Lambda_{n}\left(G,r\right)\right]=\sum_{\mathrm{dist}(\bar{\sigma },X)=r}\sum_{tr\log n=-\infty }^{-1}\\ \; & P_{G}\left(B_{\bar{\sigma }}-A_{\bar{\sigma }}=tr\log n\right)\exp\left(\beta_{n}tr\log n\right)\\ \; & \leq {\left(k-1\right)}^{r}n^{r+r\beta_{n}\left(b-a\right)/k}+\sum_{\mathrm{dist}(\bar{\sigma },X)=r}\sum_{tr\log n=r\frac{b-a}{k}\log n}^{-1}\\ \; & P_{G}\left(B_{\bar{\sigma }}-A_{\bar{\sigma }}=tr\log n\right)\exp\left(\beta_{n}tr\log n\right) \end{array}$$
$r+r\beta_{n}\left(b-a\right)/k=f_{\beta_{n}}\left(\frac{b-a}{k}\right)<\tilde{g}\left(\beta_{n}\right)$, therefore, ${\left(k-1\right)}^{r}n^{r+r\beta_{n}\left(b-a\right)/k}n^{-r\tilde{g}\left(\beta \right)/2}\leq$$n^{r(\tilde{g}\left(\beta \right)/2+\epsilon /2)}$. Using Lemma 7, we have:
$$\begin{array}{c} P_{G}\left(B_{\bar{\sigma }}-A_{\bar{\sigma }}=tr\log n\right)\exp\left(\beta_{n}rt\log n\right)\leq n^{r(f_{\beta_{n}}\left(t\right)-1+O\left(\frac{1}{\sqrt{\log n}}\right))} \end{array}$$Since $\beta_{n}\to \beta$, ∀ϵ, when *n* is sufficiently large we have $\tilde{g}\left(\beta_{n}\right)\leq \tilde{g}\left(\beta \right)+\epsilon /2$. Therefore,
$$\begin{array}{cc} \; & \sum_{\mathrm{dist}(\bar{\sigma },X)=r}\sum_{\begin{array}{c} tr\log n=\\ r(b-a)/k\log n \end{array}}^{-1}P_{G}\left(B_{\bar{\sigma }}-A_{\bar{\sigma }}=t\log n\right)\exp\left(\beta_{n}rt\log n\right)\\ \; & \leq n^{r(\tilde{g}\left(\beta_{n}\right)-\tilde{g}\left(\beta \right)/2)}\\ \; & \leq n^{r(\tilde{g}\left(\beta \right)/2+\epsilon /2)}O\left(\log n\right){\left(k-1\right)}^{r} \end{array}$$Combining the above equations, we have:
$$\begin{array}{cc} P_{G}\left(\Xi_{n}\left(r\right)\geq n^{r\tilde{g}\left(\beta \right)/2}\right) & \leq n^{r(\tilde{g}\left(\beta \right)/2+\epsilon /2)}O\left(\log n\right){\left(k-1\right)}^{r}\\ \; & \leq n^{r(\tilde{g}\left(\beta \right)/2+\epsilon )} \end{array}$$When $r>\frac{n}{\sqrt{\log n}}$, using Lemma 8, we can choose a sufficiently large constant C>1 such that $k^{n}\exp\left(-Cn\right)<e^{-n}$:
$$\begin{array}{cc} \frac{P_{\sigma |G}\left(\mathrm{dist}\left(\sigma ,X\right)=r|D\left(\sigma ,X\right)\right)}{P_{\sigma |G}\left(\sigma =X|D\left(\sigma ,X\right)\right)} & =\sum_{\begin{array}{c} D(\sigma ,X)\\ \mathrm{dist}(\sigma ,X)=r \end{array}}\frac{P_{\sigma |G}\left(\sigma =\bar{\sigma }\right)}{P_{\sigma |X}\left(\sigma =X\right)}\\ \; & >\exp(-n) \end{array}$$
happens with probability less than $e^{-n}$. Therefore, Equation (Equation 35) holds.□

If γ>b and $\beta <\beta^{*}$, we have the following lemma:

**Lemma** **10.**
*If γ>b and $\beta <\beta^{*}$, there is a set $G_{n}^{\left(1\right)}$ such that $P_{G}\left(G_{n}^{\left(1\right)}\right)\geq 1-n^{g(\bar{\beta })}$ and:*
(38)$$\begin{array}{cc} E[\sum_{r=1}^{n}\exp\left(\beta_{n}\left(A_{r}^{s}-A_{r}^{0}\right)\right)|G\in G_{n}^{\left(1\right)}] & =\left(1+o\left(1\right)\right)n^{g(\beta_{n})} \end{array}$$
(39)$$\begin{array}{cc} \mathrm{Var}[\sum_{r=1}^{n}\exp\left(\beta_{n}\left(A_{r}^{s}-A_{r}^{0}\right)\right)|G\in G_{n}^{\left(1\right)}] & \leq \left(1+o\left(1\right)\right)n^{g(2\beta_{n})} \end{array}$$


Lemma 10 is an extension of Proposition 10 in [12] and can be proved using almost the same analysis. Thus we omit the proof of Lemma 10 here.

**Lemma** **11.**
*If γ>b and $\beta <\beta^{*}$, there is a set $G_{n}^{\prime }$ such that:*
(40)$$P_{G}\left(G_{n}^{\prime }\right)\geq 1-\left(1+o\left(1\right)\right)\max\left\{n^{g(\bar{\beta })},n^{\tilde{g}\left(2\beta_{n}\right)-2g\left(\beta_{n}\right)+\epsilon }\right\}$$
*and for every $G\in G_{n}^{\prime }$,*
(41)$$\frac{P_{\sigma |G}\left(\mathrm{dist}\left(\sigma ,X\right)=1|D\left(\sigma ,X\right)\right)}{P_{\sigma |G}\left(\sigma =X|D\left(\sigma ,X\right)\right)}\geq \left(1+o\left(1\right)\right)n^{g(\beta_{n})}$$


**Proof.** The left-hand side of Equation (Equation 41) can be rewritten as:
(42)$$\frac{P_{\sigma |G}\left(\mathrm{dist}\left(\sigma ,X\right)=1\right)}{P_{\sigma |G}\left(\sigma =X\right)}=\left(1+o\left(1\right)\right)\sum_{s=1}^{k-1}\sum_{r=1}^{n}\exp\left(\beta_{n}\left(A_{r}^{s}-A_{r}^{0}\right)\right)$$Let $G_{n}^{\left(1\right)}$ be defined in Lemma 10 and $G_{n}^{\left(2\right)}:=\left\{\right|\sum_{r=1}^{n}\exp\left(\beta_{n}\left(A_{r}^{s}-A_{r}^{0}\right)\right)-\left(1+o\left(1\right)\right)n^{g(\beta_{n})}\left|\leq n^{g(\beta_{n})-\epsilon /2}\right\}$.Using Chebyshev’s inequality, we have:
$$P_{G}\left(G\notin G_{n}^{\left(2\right)}|G\in G_{n}^{\left(1\right)}\right)\leq n^{\tilde{g}\left(2\beta_{n}\right)-2g\left(\beta_{n}\right)+\epsilon }$$Let $G_{n}^{\prime }=G_{n}^{\left(1\right)}\cap G_{n}^{\left(2\right)}$:
$$\begin{array}{cc} P_{G}\left(G\in G_{n}^{\prime }\right) & =P_{G}\left(G_{n}^{\left(1\right)}\right)P_{G}\left(G\in G_{n}^{\left(2\right)}|G\in G_{n}^{\left(1\right)}\right)\\ \; & \geq \left(1-n^{\tilde{g}\left(2\beta_{n}\right)-2g\left(\beta_{n}\right)+\epsilon }\right)\left(1-n^{g(\bar{\beta })}\right)\\ \; & =1-\left(1+o\left(1\right)\right)\max\left\{n^{g(\bar{\beta })},n^{\tilde{g}\left(2\beta_{n}\right)-2g\left(\beta_{n}\right)+\epsilon }\right\} \end{array}$$
and for every $G\in G_{n}^{\prime }$,
$$\sum_{r=1}^{n}\exp\left(\beta_{n}\left(A_{r}^{s}-A_{r}^{0}\right)\right)=\left(1+o\left(1\right)\right)n^{g(\beta_{n})}$$Therefore, from Equation (Equation 42) we have:
$$\frac{P_{\sigma |G}\left(\mathrm{dist}\left(\sigma ,X\right)=1\right)}{P_{\sigma |G}\left(\sigma =X\right)}\geq \left(1+o\left(1\right)\right)n^{g(\beta_{n})}$$□

Let $\Lambda :=\{\omega^{j}\cdot 1_{n}|j=0,\ldots ,k-1\}$ where $1_{n}$ is the all-ones vector with dimension *n*, and we have the following lemma:

**Lemma** **12.**
*Suppose γ<b and $\bar{\sigma }$ satisfies $\mathrm{dist}\left(\bar{\sigma },1_{n}\right)\geq \frac{n}{\sqrt{\log n}}$ and $D(\bar{\sigma },1_{n})$. Then the event $P_{\sigma |G}\left(\sigma =\bar{\sigma }\right)>\exp\left(-Cn\right)P_{\sigma |G}\left(\sigma =1_{n}\right)$ happens with a probability (with respect to SSBM) less than $\exp(-\tau \left(\alpha ,\beta \right)n\sqrt{\log n})$ where C is an arbitrary constant, τ(α,β) is a positive number.*


**Proof.** Let $n_{r}=\left|\left\{{\bar{\sigma }}_{i}=w^{r}|i\in \left[n\right]\right\}\right|$. Then $n_{0}\geq n_{r}$ for r=1,…,k−1 since $\arg\;\min_{\sigma^{\prime }\in \Lambda }\mathrm{dist}\left(\bar{\sigma },\sigma^{\prime }\right)=1_{n}$. Without loss of generality, we suppose $n_{0}\geq n_{1}\ldots \geq n_{k-1}$. Define $N_{\bar{\sigma }}=\frac{1}{2}\left(n\left(n-1\right)-\sum_{r=0}^{k-1}n_{r}\left(n_{r}-1\right)\right)=\frac{1}{2}\left(n^{2}-\sum_{r=0}^{k-1}n_{r}^{2}\right)$. Denote the event $P_{\sigma |G}\left(\sigma =\bar{\sigma }\right)>\exp\left(-Cn\right)P_{\sigma |G}\left(\sigma =1_{n}\right)$ as $D^{\prime }\left(\bar{\sigma },C\right)$, which can be transformed as:
(43)$$\begin{array}{c} \left(1+\frac{\gamma \log n}{n}\right)\left(\sum_{{\bar{\sigma }}_{i}\neq {\bar{\sigma }}_{j},X_{i}=X_{j}}Z_{ij}+\sum_{{\bar{\sigma }}_{i}\neq {\bar{\sigma }}_{j},X_{i}\neq X_{j}}Z_{ij}\right)\\ \leq \frac{\gamma \log n}{n}N_{\bar{\sigma }}+\frac{C}{\beta }n \end{array}$$Firstly we estimate the order of $N_{\bar{\sigma }}$, and obviously $N_{\bar{\sigma }}\leq \frac{1}{2}n^{2}$. Using the conclusion in Appendix A of [18], we have:
(44)$$\sum_{r=0}^{k-1}n_{r}^{2}\leq \left\{\begin{array}{cc} nn_{0} & n_{0}\leq \frac{n}{2}\\ n^{2}-2n_{0}\left(n-n_{0}\right) & n_{0}>\frac{n}{2} \end{array}\right.$$By assumption of $\mathrm{dist}\left(\bar{\sigma },1_{n}\right)\geq \frac{n}{\sqrt{\log n}}$, we have $n_{0}\leq n-\frac{n}{\sqrt{\log n}}$ and $n_{0}\geq \frac{n}{k}$ follows from $n_{0}\geq n_{r}$. When $n_{0}>\frac{n}{2}$, we have $N_{\bar{\sigma }}\geq n_{0}\left(n-n_{0}\right)\geq \frac{n^{2}}{\sqrt{\log n}}\left(1+o\left(1\right)\right)$. The second inequality is achieved if $n_{0}=n-\frac{n}{\sqrt{\log n}}$. When $n_{0}<\frac{n}{2}$, $N_{\bar{\sigma }}\geq \frac{n^{2}-nn_{0}}{2}\geq \frac{n^{2}}{4}$ and the second inequality is achieved when $n_{0}=\frac{n}{2}$. Thus generally we have $\frac{n^{2}}{\sqrt{\log n}}\left(1+o\left(1\right)\right)\leq N_{\bar{\sigma }}\leq \frac{n^{2}}{2}$.Since $\frac{C}{\beta }n=o\left(\frac{\log n}{n}N_{\bar{\sigma }}\right)$ we can rewrite Equation (Equation 43) as:
(45)$$\left(\sum_{\begin{array}{c} {\bar{\sigma }}_{i}\neq {\bar{\sigma }}_{j}\\ X_{i}=X_{j} \end{array}}-Z_{ij}+\sum_{\begin{array}{c} {\bar{\sigma }}_{i}\neq {\bar{\sigma }}_{j}\\ X_{i}\neq X_{j} \end{array}}-Z_{ij}\right)\geq -\gamma \frac{\log nN_{\bar{\sigma }}}{n}\left(1+o\left(1\right)\right)$$Let $N_{1}=\sum_{{\bar{\sigma }}_{i}\neq {\bar{\sigma }}_{j},X_{i}=X_{j}}1$ and $N_{2}=\sum_{{\bar{\sigma }}_{i}\neq {\bar{\sigma }}_{j},X_{i}\neq X_{j}}1=N_{\bar{\sigma }}-N_{1}$.Using the Chernoff inequality we have:
$$\begin{array}{cc} P_{G}\left(D^{\prime }\left(\bar{\sigma },C\right)\right) & \leq {\left(E\left[\exp\left(-sZ\right)\right]\right)}^{N_{1}}{\left(E\left[\exp\left(-sZ^{\prime }\right)\right]\right)}^{N_{2}}\\ \; & \cdot \exp(\gamma \frac{\log nN_{\bar{\sigma }}s}{n}\left(1+o\left(1\right)\right))\\ \; & =\exp(\frac{\log n}{n}\left(1+o\left(1\right)\right)\left(e^{-s}-1\right)\left(aN_{1}+bN_{2}\right)\\ \; & +\gamma \frac{\log nN_{\bar{\sigma }}s}{n}\left(1+o\left(1\right)\right)) \end{array}$$Since s>0 and a>b, we further have:
$$\begin{array}{cc} P_{G}\left(D^{\prime }\left(\bar{\sigma },C\right)\right)\; & \leq \exp(\frac{N_{\bar{\sigma }}\log n}{n}\left(b\left(e^{-s}-1\right)+\gamma s+o\left(1\right)\right)) \end{array}$$Let $h_{b}\left(x\right)=x-b-x\log\frac{x}{b}$, which satisfies $h_{b}\left(x\right)<0$ for 0<x<b, and take $s=-\log\frac{\gamma }{b}>0$, using $N_{\bar{\sigma }}\geq \frac{n^{2}}{\sqrt{\log n}}$ we have:
$$\begin{array}{cc} P_{G}\left(D^{\prime }\left(\bar{\sigma },C\right)\right) & \leq \exp(N_{\bar{\sigma }}\frac{\log n}{n}h_{b}\left(\gamma \right)\left(1+o\left(1\right)\right))\\ \; & \leq \exp(h_{b}\left(\gamma \right)n\sqrt{\log n}\left(1+o\left(1\right)\right)) \end{array}$$□

**Proof of Theorem** **2.**(1) Since $P_{\sigma }\left(\sigma \notin S_{k}\left(X\right)\right)=\sum_{f\in S_{k}}P_{\sigma }\left(\sigma \neq f\left(X\right)|D\left(\sigma ,f\left(X\right)\right)\right)P_{\sigma }\left(D\left(\sigma ,f\left(X\right)\right)\right)$, we only need to establish $P_{\sigma }\left(\sigma \neq X|D\left(\sigma ,X\right)\right)\leq n^{\tilde{g}\left(\beta \right)/2+\epsilon }$. From Lemma 9, we can find $G_{n}^{\left(r\right)}$ for r=1,…,n. Let $G_{n}^{\prime }=\cap_{r=1}^{n}G_{n}^{\left(r\right)}$ and choose $\frac{\epsilon }{2}$; from Equations (Equation 32) and (Equation 34), we have:
$$\begin{array}{cc} P_{G}\left(G_{n}^{\prime c}\right) & =P_{G}\left(\cup_{r=1}^{n}{\left(G_{n}^{\left(r\right)}\right)}^{c}\right)\\ \; & \leq \sum_{r=1}^{n/\sqrt{\log n}}n^{r(\tilde{g}\left(\beta \right)/2+\epsilon /2)}+ne^{-n}\\ \; & \leq \frac{1}{2}n^{\tilde{g}\left(\beta \right)/2+\epsilon } \end{array}$$
where the last equality follows from the estimation of sum of geometric series. On the other hand, for every $G\in G_{n}^{\prime }$, from Equations (Equation 33) and (Equation 35), we have:
$$\begin{array}{cc} \frac{P_{\sigma |G}\left(\sigma \neq X|D\left(\sigma ,X\right)\right)}{1-P_{\sigma |G}\left(\sigma \neq X|D\left(\sigma ,X\right)\right)} & =\frac{P_{\sigma |G}\left(\sigma \neq X|D\left(\sigma ,X\right)\right)}{P_{\sigma |G}\left(\sigma =X|D\left(\sigma ,X\right)\right)}\\ \; & <\sum_{r=1}^{n/\sqrt{\log n}}n^{r\tilde{g}\left(\beta \right)/2}+ne^{-n} \end{array}$$
from which we can get the estimation $P_{\sigma |G}\left(\sigma \neq X|D\left(\sigma ,X\right)\right)\leq \frac{1}{2}n^{\tilde{g}\left(\beta \right)/2+\epsilon }$. Finally,
$$\begin{array}{cc} P_{\sigma }\left(\sigma \neq X|D\left(\sigma ,X\right)\right) & =\sum_{G\in G_{n}^{\prime }}P_{G}\left(G\right)P_{\sigma |G}\left(\sigma \neq X|D\left(\sigma ,X\right)\right)\\ +P_{G}\left(G_{n}^{\prime c}\right)\; & \leq n^{\tilde{g}\left(\beta \right)/2+\epsilon }. \end{array}$$(2) When $\beta <\beta^{*}$, using Lemma 11, for every $G\in G_{n}^{\prime }$ we can obtain:
$$\frac{1-P_{\sigma |G}\left(\sigma =X|D\left(\sigma ,X\right)\right)}{P_{\sigma |G}\left(\sigma =X|D\left(\sigma ,X\right)\right)}\geq \left(1+o\left(1\right)\right)n^{g(\beta_{n})}$$We then have:
$$P_{\sigma |G}\left(\sigma =X|D\left(\sigma ,X\right)\right)\leq \left(1+o\left(1\right)\right)n^{-g(\beta_{n})}$$Then:
$$\begin{array}{cc} \; & P_{\sigma }\left(\sigma =X|D\left(\sigma ,X\right)\right)\leq P\left(G_{n}^{\prime c}\right)\\ \; & +\sum_{G\in G_{n}^{\prime }}P_{G}\left(G\right)P_{\sigma |G}\left(\sigma =X|D\left(\sigma ,X\right)\right)\\ \; & \leq \left(1+o\left(1\right)\right)n^{-g(\beta_{n})}+\left(1+o\left(1\right)\right)\max\left\{n^{g(\bar{\beta })},n^{\tilde{g}\left(2\beta_{n}\right)-2g\left(\beta_{n}\right)+\epsilon }\right\}\\ \; & \leq \left(1+o\left(1\right)\right)\max\left\{n^{g(\bar{\beta })},n^{-g(\beta )+\epsilon }\right\} \end{array}$$(3) When γ<b, for any $f\in S_{k}$, we have $\mathrm{dist}\left(f\left(X\right),\Lambda \right)=\frac{(k-1)n}{k}>\frac{n}{\sqrt{\log n}}$. Therefore, using Lemma 12, we can find a graph $G_{n}^{\prime }$ such that $P_{C}\left(G_{n}^{\prime }\right)\leq \exp\left(-nC\right)$ and for any $G\notin G_{n}^{\prime }$, $P_{\sigma |G}\left(\sigma =f\left(X\right)\right)\leq \exp\left(-Cn\right)$. Therefore,
$$\begin{array}{c} P_{a}\left({\hat{X}}^{*}\right)\leq P_{G}\left(G_{n}^{\prime }\right)+k!\exp\left(-Cn\right)=\left(1+k!\right)\exp\left(-Cn\right) \end{array}$$The conclusion of $P_{a}\left({\hat{X}}^{*}\right)\leq \exp\left(-Cn\right)$ follows since *C* can take any positive value. □

### 8.3. Proof of Theorem 3

**Lemma** **13**(Lemma 8 of [7])**.**
*Let m be a positive number larger than n. When $Z_{1},\ldots ,Z_{m}$ are i.i.d. Bernoulli($\frac{b\log n}{n}$) and $W_{1},\ldots ,W_{m}$ are i.i.d Bernoulli($\frac{a\log n}{n}$), independent of $Z_{1},\ldots ,Z_{m}$, then:*
(46)$$P\left(\sum_{i=1}^{m}\left(Z_{i}-W_{i}\right)\geq 0\right)\leq \exp\left(-\frac{m\log n}{n}{\left(\sqrt{a}-\sqrt{b}\right)}^{2}\right)$$

**Proof of Theorem** **3.**Let $P_{F}^{\left(r\right)}$ denote the probability when there is σ satisfying dist(σ,X)=r and H(σ)<H(X).From Equation (Equation 23), when σ differs from *X* only by one coordinate, from Lemma 13 the probability for H(σ)<H(X) is bounded by $P_{G}\left(A_{r}^{s}-A_{r}^{0}>0\right)\leq n^{-\frac{{\left(\sqrt{a}-\sqrt{b}\right)}^{2}}{k}}$. Therefore, $P_{F}^{\left(1\right)}\leq \left(k-1\right)n^{g(\bar{\beta })}$. Using Lemma 7, we can get $P_{F}^{\left(r\right)}\leq {\left(k-1\right)}^{r}n^{rg(\bar{\beta })}$ for $r\leq \frac{n}{\sqrt{\log n}}$. For $r\geq \frac{n}{\sqrt{\log n}}$, choosing C=0 in Lemma 8 we can get $\sum_{r\geq \frac{n}{\sqrt{\log n}}}P_{F}^{\left(r\right)}\leq e^{-n}$. That is, the dominant term is $\sum_{r\leq \frac{n}{\sqrt{\log n}}}P_{F}^{\left(r\right)}$ since the other part decreases exponentially fast. Therefore, the upper bound for error rate of ${\hat{X}}^{\prime }$ is:
$$\begin{array}{cc} P_{F}= & \sum_{r=1}^{n}P_{F}^{\left(r\right)}\leq \left(1+o\left(1\right)\right)\sum_{r=1}^{\infty }{\left(k-1\right)}^{r}n^{rg(\bar{\beta })}\\ \leq & \left(1+o\left(1\right)\right)\frac{\left(k-1\right)n^{g(\bar{\beta })}}{1-\left(k-1\right)n^{g(\bar{\beta })}}=\left(k-1+o\left(1\right)\right)n^{g(\bar{\beta })} \end{array}$$When $\sigma \in W^{*}$, since $|\left\{r\in \left[n\right]|\sigma_{r}=\omega^{i}\right\}|=I_{i}^{\prime }+\frac{n}{k}-I_{i}$, we have $I_{i}^{\prime }=I_{i}$. From Lemma 6, we can see $|B_{\bar{\sigma }}\left|=\right|A_{\bar{\sigma }}|$ and $N_{\bar{\sigma }}=0$. Then from Equation (Equation 24), $H\left(\bar{\sigma }\right)<H\left(X\right)$ is equivalent with $B_{\bar{\sigma }}>A_{\bar{\sigma }}$.Therefore, when $\mathrm{dist}\left(\bar{\sigma },X\right)\geq \frac{n}{\sqrt{\log n}}$ and D(σ,X), from Equation (Equation 31), we have $A_{\bar{\sigma }}\geq \frac{n^{2}}{k^{2}\left(k-1\right)\sqrt{\log n}}\left(1+o\left(1\right)\right)$. We use Lemma 13 by letting $m=|A_{\bar{\sigma }}|$ when $m\geq \frac{n}{\sqrt{\log n}}$; the error term is bounded by $\sum_{r\geq \frac{n}{\sqrt{\log n}}}P_{F}^{\left(r\right)}\leq \sum_{r\geq \frac{n}{\sqrt{\log n}}}{\left(k-1\right)}^{r}\exp\left(-\frac{{\left(\sqrt{a}-\sqrt{b}\right)}^{2}}{k^{2}\left(k-1\right)}n\sqrt{\log n}\right))\leq \exp\left(-n\right)$, which decreases exponentially fast. For $m<\frac{n}{\sqrt{\log n}}$, we can use Lemma 7 directly by considering $\sum_{r=2}^{\frac{n}{\sqrt{\log n}}}P_{F}^{\left(r\right)}$. The summation starts from r=2 since $\sigma \in W^{*}$. Therefore,
$$\begin{array}{cc} P_{F}= & \sum_{r=2}^{n}P_{F}^{\left(r\right)}\leq \left(1+o\left(1\right)\right)\sum_{r=2}^{\infty }{\left(k-1\right)}^{r}n^{rg(\bar{\beta })}\\ \leq & \left({\left(k-1\right)}^{2}+o\left(1\right)\right)n^{2g(\bar{\beta })}. \end{array}$$□

## Figures and Tables

**Figure 1 entropy-23-00065-f001:**
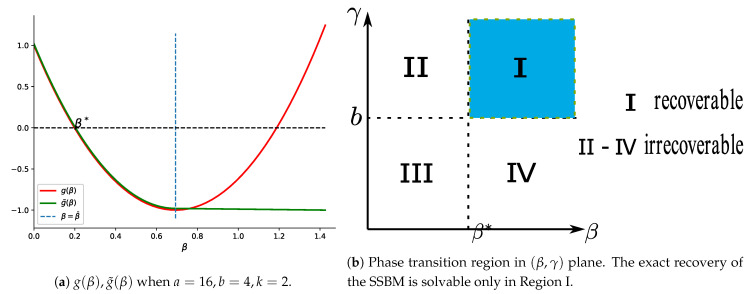
Illustration of Theorem 2.

**Figure 2 entropy-23-00065-f002:**
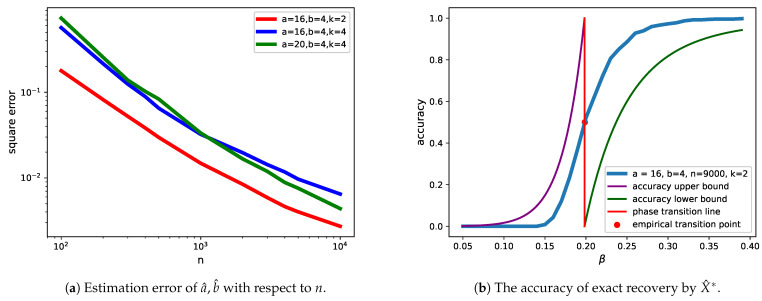
Experimental results.

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
