# Peer review of "Exact Recovery of Stochastic Block Model by Ising Model"

_entropy, 2021, doi:10.3390/e23010065_

Round 1

Reviewer 1 Report

Please improve description of "community detection" and the meaning of "exact recovery". Unless one already works in this field it is not easy to understand from the current introduction.

My main problem with the paper is the language, and probably som minor errors, which makes it hard to follow parts of the ms. For example, definition 2 seems to be missing an essential part: "If there exists...such that" but there is no "then" part to describe what is defined. 

On line 109 the authors mention the "ground truth label vector", what does this mean? Is X always a ground truth label vector? 

The figure on page 4 is missing the powers of 10 on the y-axis.

On line 131-132 the use of repulsive and attractive looks strange, should it be repellent (or repelling) and attracting? On line 217 and in Algorithm 1, what does disassortative mean?

At the bottom of page 6 the "almost 1" should perhaps be "almost surely"?

Lemma 2: Umlauts missing on Erdos.

Lemma 6: The ending of the first sentence looks strange, perhaps from defining |I| and using it at once. Split it up?

Sentence at top of page 14: "Under such condition..." has a period after the formula, should probably be a comma.

Reviewer 2 Report

In their work, the Authors study the phase transition of an Ising model defined on a special random graph, called "stochastic block model". As explicitly stated by the Authors in page 2, one of the goals of the present work is to "extend the connection of SBM and Ising model to multiple community case, establish the phase transition property and give the recovery error upper bound". In this respect, I found the mention to phase transitions in the Ising model, in the first line of page 2, quite loose and incomplete. On the one hand, it is in fact well known that the 2D Ising model, in the standard equilibrium statistical mechanical setting, undergoes either a second order phase transition when varying the temperature (e.g. at zero external magnetic field) or a first order one when varying the magnetic field (at low temperature). On the other hand, recent results indicate that the 2D Ising model may also undergo a non-equilibrium phase transition, characterized by the onset of the so-called uphill currents, when the spin system is coupled at the boundaries to external magnetization reservoirs. See e.g.:

  1. M. Colangeli, C. Giardinà, C. Giberti, C. Vernia, Nonequilibrium two-dimensional Ising model with stationary uphill diffusion, Phys. Rev. E 97, 030103(R) (2018);

  2. M. Colangeli, C. Giberti, C. Vernia, M. Kröger, Emergence of stationary uphill currents in 2D Ising models: the role of reservoirs and boundary conditions, Eur. Phys. J. Special Topics 228, 69-91 (2019).

Thus, in order to strengthen the theory discussed in the present manuscript, I believe that, for the sake of completeness, it would be relevant to check whether the formalism developed in this work may also tackle, in principle, the onset of non-equilibrium phase transitions of the Ising model as discussed in the two references listed above. 

Reviewer 3 Report

The authors of this paper study the exact solution of the Stochastic Block Model (SBM) for several communities making use of an Ising model formulation of the problem. The problem is interesting itself from the mathematical point of view and also the multiple applications that the SBM has for generation of (modular) complex networks. I didn't notice any major flaw in the calculations, though I would insist on the presentation. -- In fact, the paper is presented as a very classic mathematical paper, making the accessibility of the Entropy readers hard. I would suggest a better organization of it (maybe putting some of the proofs of lemmas in the Appendix). -- Also my next point is related to the fact that the paper is an extension of a very recent preprint for the case of two modules (from one of the authors). This should be clearly stated to distinguish the novelty of the actual paper.

Round 2

Reviewer 2 Report

In their reply, the Authors completely eluded the questions raised in my report. In particular, they state the following: "In you comment, you mentioned three cases for 2D Ising model. I think my study is more similar with the standard equilibrium statistical mechanics setting when no external magnetic field is present and the phase transition occurs by varying the temperature". 

I would like point out that in the 2D Ising model referred to in my report there is no external magnetic field either, and the phase transition may indeed occur by only varying the temperature. Hence, the set-up of the aforementioned 2D Ising model is certainly close to that considered in the present work. In fact, the aim of my question was to strengthen the results of the present manuscript by highlighting possible analogies with similar models reported in the literature. In this respect, I believe that the amended version of the manuscript does not shed any new light on the theory of phase transitions for Ising-like models, hence I find it unsuitable for publication on the Journal Entropy.